# Effect of *Saccharomyces cerevisiae* Supplementation on Reproductive Performance and Ruminal Digestibility of Queue Fine de l'Ouest Adult Rams Fed a Wheat Straw-Based Diet

Samia Ben Saïd [1,*], Jihene Jabri [2], Sihem Amiri [1], Mohamed Aroua [1], Amel Najjar [3], Sana Khaldi [2], Zied Maalaoui [4], Mohamed Kammoun [2] and Mokhtar Mahouachi [1]

1 Ecole Supérieure d'Agriculture du Kef, Université de Jendouba, Jendouba 8189, Tunisia
2 Ecole Nationale de Médecine Vétérinaire, Université de Mannouba, Mannouba 2010, Tunisia
3 Institut Nationale Agronomique de Tunisie, Université de Carthage, Carthage 1054, Tunisia
4 Arm and Hammer Animal and Food Production, Ewing, NJ 08628, USA
* Correspondence: bensaid.samia@esakef.u-jendouba.tn

**Abstract:** This study aimed to investigate the effect of supplementing a wheat straw-based diet with *Saccharomyces cerevisiae* (S.C.) on feed intake, nutrient digestibility, nitrogen balance, body weight and reproduction performance. The experiment was conducted on 14 Queue Fine de l'Ouest rams between 3 and 4 years of age (body weight (B.W.): $54.7 \pm 2.03$ kg; body condition score (B.C.S.): $3.5 \pm 0.5$), for 80 days during the breeding season. The rams were divided into two homogenous groups (n = 7), housed individually in floor pens, and allocated to two dietary treatments. The control group was offered a basal diet of 1 kg/d of wheat straw and 700 g of concentrate. The experimental group (yeast) received the same basal diet supplemented with 10 g of S.C./head/day. The results indicated that the S.C. supplementation had no significant effect on the animal's body weight, volume and concentration of semen, dry matter intake, crude protein digestibility and nitrogen balance. Compared to the control group, the S.C. addition improved ($p < 0.05$) the digestibility of dry matter by 7.3%, organic matter by 11.9% and crude fiber by 24%. In addition, the mass motility score increased for the yeast group compared to the control ($3.7 \pm 0.24$ vs. $1.9 \pm 0.27$, $p < 0.05$). The total number of dead and abnormal spermatozoa decreased for the yeast group in contrast to the control group ($9.28 \pm 0.95$ vs. $26.8 \pm 3.85\%$ and $25.5 \pm 3.33$ vs. $59.2 \pm 2.78\%$, respectively; $p < 0.05$). These results showed that adding S.C. to Queue Fine de l'Ouest ram's diet during breeding season could improve nutrient digestibility and reproductive performance.

**Keywords:** ram; sperm quality; *Saccharomyces cerevisiae*; apparent digestibility

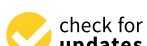



## 1. Introduction

Animal feeding is vital for maintaining animal health, including reproductive performance, and producing safe animal products [1]. Puberty onset, ovulation rate, embryo survival, depth of anestrus and the response to the male effect were all proven to be affected by dietary changes in ewes [2]. High-energy diets increase the libido, testes size, seminiferous tubule diameter and spermatogenesis for rams [3] and promote the onset of puberty in young males [4,5]. Furthermore, malnutrition was reported to reduce the scrotal circumference and inhibit the intensity of the goat male's sexual behavior [6].

Despite their poor digestibility and their negative impact on animal productivity [7], forages and agricultural by-products continue to be the primary feeding source for ruminants.

This has created an interest among nutritionists in manipulating the rumen ecosystem and fermentation characteristics of livestock in order to improve feed utilization, animal productivity and health [8,9].

Currently, probiotics that are live and non-pathogenic microbes, such as live yeast (*Saccharomyces cerevisiae*), are widely used as feed additives for ruminants. Probiotic supplementation has proved its effectiveness for enhancing intestinal function by maintaining

a healthy gastrointestinal environment, thus improving nutrient intake and digestibility, production performance and feed efficiency [10–13]. Further, Saccharomyces cerevisiae (S.C.) application to the ruminant diet has been shown to reduce methane emissions from anaerobic fermentation [14], decrease the number of pathogenic intestinal bacteria [15] and repopulate gut microflora in diarrheal cases [16]. However, far too little attention has been paid to the effect of S.C. supplementation on ruminant's reproductive performance. Recently, Ahmadzadeh [17] proved that S.C. supplementation to the ewe in the breeding season could enhance the reproductive performance by increasing fertility and the twinning rate. Hence, this study aimed to evaluate the effect of S.C. supplementation on reproductive performance, intake and ruminal digestibility of native Tunisian rams fed a wheat straw and concentrate diet during the breeding season.

## 2. Materials and Methods

The study was carried out on the experimental farm of the Higher School of Agricultural of Kef, located in the northwestern part and the semi-arid area of Tunisia (latitude 35.7° N, longitude 8.83° W). The experiment meets the ethical guidelines and adheres to Tunisian legal requirements (The Livestock Law No. 2005-95 of 18 October 2005) and was conducted by trained staff strictly following good animal practices as defined by national authorities.

### 2.1. Experimental Design

A total of 14 Queue Fine de l'Ouest rams (also called "Bergui" in the local language) were used during the traditional mating season in Tunisia for 80 days from March to May. Animals were aged between 3 and 4 years, with a mean body weight of $54.75 \pm 2.03$ kg and a mean body condition score (B.C.S.) of $3.5 \pm 0.5$. The rams were checked for the integrity of their sexual tracts before the experiment, then assigned into two homogenous groups. For the control group (n = 7), rams received 1 kg of wheat straw and a fixed amount of concentrate (700 g) according to their metabolic weight. The concentrate was composed of 9% soyabean meal, 87% barley and a 4% mineral and vitamin complement. For the experimental group (yeast, n = 7), the same diet was supplemented with 10 g/head/day of Saccharomyces cerevisiae yeast culture Celmanax® (Arm & Hammer Animal and Food Production, Ewing, NJ, USA). The chemical composition of the diet ingredients (concentrate and wheat straw) is detailed in Table 1. Animals were allowed free access to fresh and clean water and received two equal meals at 09:00 a.m. and 15:00 p.m. The rams were housed individually in cleaned experimental pens of 2 m² (1 m × 2 m) and raised under the same management and nutritional conditions. They were exposed to the same normal seasonal photoperiod without any contact with ewes. All rams were vaccinated against enterotoxemia (Coglavax®, CEVA, Libourne, France, 2 mL/animal) and received an antiparasitic treatment (IVOMEC®, Boehringer-Ingelheim, Reims, France, 1 mL/50 kg B.W. by subcutaneous rout).

**Table 1.** Ingredients of the experimental diets and their chemical composition (g/kg fresh matter).

|  | Dry Matter | Mineral | Crude Protein | Crude Fiber |
|---|---|---|---|---|
| Wheat straw | 890 | 54 | 42 | 430 |
| Concentrate | 910 | 50 | 182.9 | 70 |

### 2.2. Measurements

#### 2.2.1. Feed Intake, Live Weight, Body Score Condition and Testicular Diameter

The daily feed intake of concentrate and wheat straw (the difference between offered feeds and refusals) for each animal was recorded throughout the trial. Samples of feed offered and refused were removed and weighed before every morning feeding and pooled over the week for each ram. Live body weight, body score condition (B.S.C., noted on a scale from 0 to 5, [18]) and testicular diameter were measured every two weeks before

distributing the morning's food. The testicular diameter was assessed using a clipper with the ram in a standing position.

### 2.2.2. Semen Collection and Evaluation

Sperm collection was performed every 10 days. Rams were put individually in the collection room in the presence of a teaser female that was previously induced by inserting a progestogen-impregnated vaginal sponge for 10 days [19]. After collection, sperm volume was immediately recorded by direct reading from a graduated glass tube. The general appearance of the semen was visually assessed [20]. Samples were immediately placed in a water bath at 35 °C [19]. Mass activity (wave motion or motility score) in undiluted semen and individual motility in diluted semen were assessed by examining a drop of semen under a warm stage using a phase-contrast microscope (score, 0–5, [21]). The sperm concentration was assessed by a hemocytometer slide (Malassez slide, Marienfeld, Germany) [21]. Percentages of dead and abnormal spermatozoa were studied using the eosin/nigrosin staining technique described by Baril et al. [21].

### 2.2.3. Digestibility Trial

Over the last two weeks of the trial, rams were placed in an individual metallic box with a wire-mesh floor (1.2 m × 0.6 m) to evaluate the diet's (in vivo) digestibility and nitrogen balance (7 days for adaptation and 7 days for measurements) [22]. The crates were specifically designed to separate feces, urine and refusals. The offered feed, refusals, total urine and feces excreted during 24 h by each ram were collected and weighed daily before the morning feeding. A sample of 10% of raw material from the feed refusals and total feces of each ram was pooled daily (during the 7 days of measurements). The fecal samples were stored at −20 °C until further analysis. The total urine was collected daily into plastic buckets and preserved with 50 mL of 10% sulfuric acid (0.1 N) solution. Representative aliquots of each animal's urine (10%) were immediately stored in a freezer (−20 °C) until analysis.

For each ram, samples of offered diets, refusals and feces were dried in a forced air oven at 55 °C until a constant weight [23]. The samples were subsequently ground through a 1 mm screen using a Wiley mill. Dry matter, ash, crude fiber and crude protein contents were determined according to the Association of Official Analytical Chemists [23]. Moreover, the nitrogen content of urine was determined according to the Kjeldahl method [23]. All chemical analyses were performed in triplicate for each sample.

### 2.3. Statistical Analysis

Mixed models with a random animal effect were run for repeated data on body weight, testicular diameter and semen parameters (volume, concentration, massal motility and individual motility, percentages of dead and morphologically abnormal spermatozoa) using the MIXED model's procedure (S.A.S. Version 9.1; S.A.S. Inst. Inc., Cary, NC, USA). We included the fixed effect of the treatment, the rank of the sperm collection (week of collection) and their interaction in the analysis. The random variable was the ram within the treatment. An ANOVA was performed to study the effect of treatment on dry matter, digestibility parameters and nitrogen balance. Duncan's test was used to compare variables between groups. Data on dry matter intake, digestibility parameters and nitrogen balance were compared using the analysis of variance (ANOVA) [24]. Each ram was regarded as the experimental unit. The significance level was set at 0.05, and trends were discussed for $p$-values between 0.05 and 0.10.

## 3. Results

### 3.1. Live Body Weight, Body Condition Score and Testicular Diameter

The live body weight and B.C.S. of rams did not vary between the experimental and control groups ($p > 0.05$; Table 2). Regarding the testicular diameter, results showed no variation between the treated and the control groups ($p > 0.05$).

**Table 2.** Live weight, body score condition and testicular diameter of control and yeast-treated rams (means ± S.E.M.).

| | Beginning * | | End * | |
|---|---|---|---|---|
| | **Control** | **Yeast** | **Control** | **Yeast** |
| L.B.W. (kg) | 55.7 ± 4.0 | 54.8 ± 3.8 | 56.0 ± 1.9 | 55.05 ± 2.7 |
| B.S.C. | 3.1 ± 0.32 | 3.2 ± 0.27 | 3.4 ± 0.21 | 3.5 ± 0.15 |
| T.D. (cm) | 4.95 ± 0.20 | 4.90 ± 0.31 | 4.93 ± 0.18 | 4.83 ± 0.12 |

S.E.M: standard error of the mean; L.B.W.: live body weight, B.S.C.: body score condition, T.D.: testicular diameter. * Beginning: mean data relative to the first sample collection during the trial. End: mean data of the last sample collection during the trial.

### 3.2. Semen Characteristics

Data for semen characteristics such as ejaculation volume, sperm concentration, motility scores and percentage of dead and abnormal spermatozoa are represented in Figure 1. There was no significant difference between the two groups regarding the ejaculation volume, mass motility, individual motility, sperm concentration and the percentage of abnormal spermatozoa at the beginning of the trial (Table 3; $p > 0.05$).

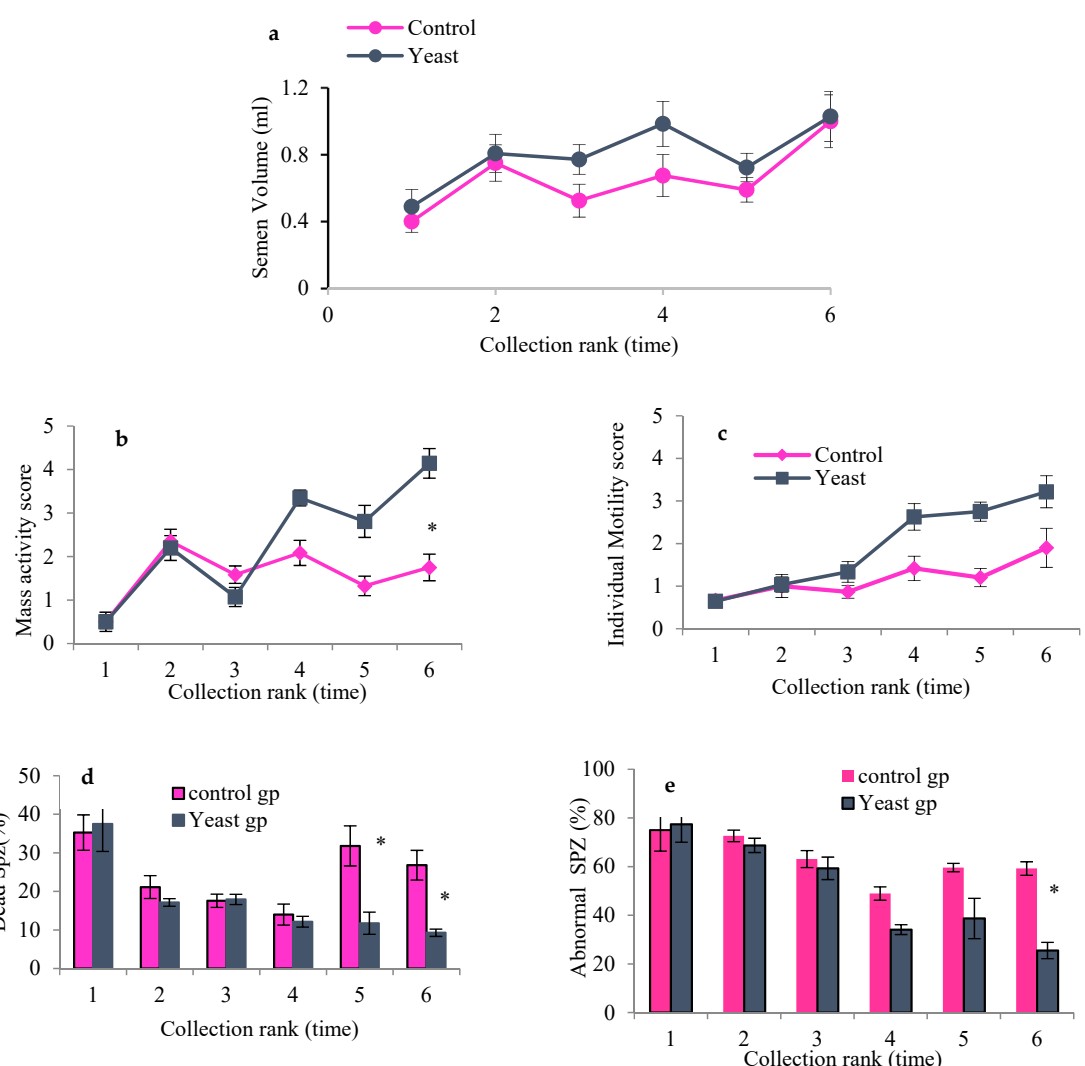

**Figure 1.** Variation in (**a**) ejaculate volume, (**b**) mass motility score, (**c**) individual motility, (**d**) dead spermatozoa rate and (**e**) abnormal spermatozoa rate in yeast-treated and control rams. Effect of supplementation is significant, * $p < 0.05$.

**Table 3.** Sperm quality traits of control and yeast-treated Queue Fine de l'Ouest rams (means ± S.E.M.).

| | Beginning * | | End * | |
|---|---|---|---|---|
| | Control | Yeast | Control | Yeast |
| Volume (mL) | 0.4 ± 0.06 | 0.48 ± 0.1 | 1 ± 0.15 | 1.02 ± 0.14 |
| MM | 0.77 ± 0.12 | 0.75 ± 0.18 | 1.91 ± 0.27 [a] | 3.70 ± 0.24 [b] |
| MI | 0.66 ± 0.08 | 0.64 ± 0.14 | 1.90 ± 0.40 | 3.20 ± 0.30 |
| CC ($10^6$ spz/mL) | 1512.29 ± 86.91 | 1840.7 ± 246 | 2036.47 ± 127 [a] | 2296.23 ± 132 [b] |
| Dead spz (%) | 35.25 ± 4.59 | 37.5 ± 7.13 | 26.8 ± 3.85 [a] | 9.28 ± 0.95 [b] |
| Abnormal spz (%) | 75 ± 8.62 | 77.4 ± 7.38 | 59.2 ± 2.78 [a] | 25.5 ± 3.33 [b] |

MM: mass motility score; MI: individual motility score, CC: sperm concentration, spz: spermatozoa. a,b: different letters within the same row (different diets) differ significantly ($p < 0.05$). S.E.M.: standard error of the mean. * Beginning: mean data relative to the first sample collection during the trial. End: mean data of the last sample collection during the trial.

Regardless of the treatment, the mean ejaculation volume was 0.73 ± 0.2 mL (0.66 ± 0.2 and 0.80 ± 0.1 for control and yeast-treated rams, respectively, $p > 0.05$; Table 3). However, the ejaculate volume was significantly affected by the rank of sperm collection ($p < 0.05$). The volume increased during the weeks of the trial (Figure 1a) and the sixth sperm collection showed the highest sperm volume (v = 1 mL, $p < 0.05$). The interaction between the treatment and sperm collection had no significant effect on sperm volume ($p > 0.05$). At the end of the experiment, S.C. supplementation improved the mass motility (Table 3) compared with the control group (3.7 ± 0.24 vs. 1.91 ± 0.17, $p < 0.05$). Statistical analysis also showed a significant effect on the collection rank ($p < 0.05$) and on the interaction between the treatment and the collection rank ($p < 0.05$; Figure 1b). In the same way, statistical analysis showed the effect of the collection rank ($p < 0.05$) on individual motility. However, there was no effect of S.C. supplementation on this parameter (Figure 1c).

S.C. supplementation slightly improved sperm concentration ($p = 0.056$; Table 3). Mean sperm concentration during the whole period of the experiment was 2240 ± 201 × $10^6$ spz/mL and 2534 ± 190 × $10^6$ spz/mL for control and yeast-treated rams, respectively. In addition, an effect of the collection rank was observed ($p < 0.05$).

The percentage of dead spermatozoa decreased ($p < 0.05$) during the trial (Table 3 and Figure 1d). At the end of the experiment, the improvement in this parameter was more pronounced ($p < 0.05$) with the S.C. diet than in the control one (9.28 ± 0.95% and 26.8 ± 3.85, respectively). The interaction between the treatment and the collection rank was significant ($p < 0.05$).

The mean of the abnormal spermatozoa percentage followed the same trend as the dead spermatozoa since it decreased significantly ($p < 0.05$) during the trial for both diets. However, this decrease was more significant in the yeast-supplemented group than in the control group (59.2 ± 2.78 vs. 25.5 ± 3.33%, respectively, $p < 0.05$; Table 3).

### 3.3. Nutrient Intake and Diet Digestibility

The results revealed no significant effect ($p > 0.05$) of the diet supplementation with S.C. on the daily feed intake (g $kg^{-1}$ body weight per day) (Table 4). However, for the nutrient's digestibility (Table 5), the addition of S.C. to the rams' diet improved the dry matter, organic matter and fiber digestibility by 7.3% ($p = 0.02$), 11.9% ($p = 0.01$) and 24% ($p = 0.02$), respectively, as compared with the control. Likewise, protein digestibility tended to increase by 10.5% ($p = 0.08$). Moreover, the N intake, excreted nitrogen (urinary nitrogen (U.R.) and fecal nitrogen (F.N.)) and nitrogen retention (Table 5) were unaffected by the addition of S.C. to the diet ($p > 0.05$).

**Table 4.** Effect of adding S.C. supplementation on nutrient intake (g/kg B.W.$^{0.75}$/day) of rams (means ± S.E.M.).

|  | Control | Yeast |
|---|---|---|
| Dry matter | 63.3 ± 1.00 | 68.5 ± 1.10 |
| Crude protein | 6.4 ± 0.04 | 6.8 ± 0.11 |
| Crude fiber | 14.4 ± 0.38 | 15.6 ± 0.31 |
| Organic matter | 60.3± 0.95 | 65.3 ± 1.05 |

S.E.M.: standard error of the mean.

**Table 5.** Apparent digestibility (%) and nitrogen balance (g/d) parameters (means ± S.E.M.).

|  | Control | Yeast |
|---|---|---|
| Dry matter digestibility (D.M.D.) | 70.19 ± 0.90 [a] | 75.30 ± 0.96 [b] |
| Organic matter digestibility (O.M.D.) | 58.73 ± 1.22 [a] | 65.72 ± 1.25 [b] |
| Crude protein digestibility (C.P.D.) | 62.07 ± 1.20 | 68.61 ± 2.06 |
| Crude fiber digestibility (C.F.D.) | 40.44 ± 2.51 [a] | 50.10 ± 1.61 [b] |
| Nitrogen intake (g/d) | 22.45 ± 0.91 | 21.72 ± 0.77 |
| Fecal nitrogen (g/d) | 13.32 ± 0.48 | 12.4 ± 0.74 |
| Urinary nitrogen (g/d) | 4.22 ± 0.28 | 5.36 ± 0.80 |
| Retained nitrogen (g/d) | 4.91 ± 0.86 | 3.96 ± 0.73 |

a,b: different letters within the same row (different diets) differ significantly ($p < 0.05$). S.E.M.: standard error of the mean.

## 4. Discussion

In this study, adding 10 g/day of S.C. during the experimental period did not affect body live weight, testicular diameter, ejaculation volume and concentration. This may be explained, in part, by the fact that there was not any measured difference in the feed intake between both dietary groups. In addition, the animals are already mature, thus they do not have additional needs. Although there is a general agreement that prebiotic supplementation to ruminant feeding increases the feed intake [25], the S.C. addition to the rams' diet did not affect feed intakes for dry matter, crude proteins, crude fiber and organic matter in our study. In agreement with our finding, Haddad and Goussous [26] and Titi et al. [27] reported that supplying yeast culture at increasing addition rates of 0, 3, 6 and 12.6 g/day to Awassi lambs' diets had no significant effect on nutrient intake. In another meta-analysis of 141 yeast treatment studies, Desnoyers et al. [28] reported that the effect of yeast supplementation on ruminant diets is related to some interfering factors. Indeed, the positive effect on intake could be affected by the proportion of concentrate in the diet but not be influenced by the neutral detergent fiber or ruminant species. Issakowicz et al. [29] reported a significant increase in dry matter intake and the feed conversion ratio by adding S.C. to lamb-fed diets with a greater concentrate proportion (80%).

Several studies reported the interrelationship between energy intake and reproductive performance in adult rams [30–32]. In this regard, Tufarelli et al. [33] reported that a dietary level with a higher concentrate of supplementation would improve body weight gain, feed intake and semen characteristics (volume and concentration) of rams. In the Tunisian Queue Fine de l'Ouest breed, alternating feeding levels from high (1.6 of the metabolizable energy for maintenance (M.E.M.) requirement) to low (1.0 M.E.M.) levels was associated with a reduction in the volume of ejaculate, but did not reduce the sperm concentration in rams [9]. The current study showed that both quantitative and qualitative parameters of rams' semen were improved during the trial period independently of the diet. These results are similar to those reported by Mahouachi and Khaldi [34], suggesting a resumption of sexual activity of the local breed at the beginning of the traditional mating season even if the photoperiod is unfavorable. Therefore, there would be interference between the two factors of diet and time, as shown by the significant interactions. Compared to the control group, S.C. supplementation had no apparent effect on ejaculation volume. However, it increased the mass motility significantly after 60 days of its introduction to the diet.

Furthermore, the percentage of dead spermatozoa and abnormal spermatozoa were reduced with the yeast supplementation compared to the control group. These results suggest that yeast supplementation's main positive effect is improving sperm quality. In this regard, Aboul-Ela et al. [35] reported that S.C. reduces primary sperm abnormalities in rats. Furthermore, Emmanuel et al. [36] showed that an S.C.-based diet in bucks improved motility and live sperm, tubule diameter, epididymal volume, the volume fraction of the duct and total duct volume. They also reported that the incidence of head and tail sperm abnormalities decreased significantly in bucks fed S.C.-based diets compared with the control diet.

For humans, Sahar-Eissa et al. [37] found that the baker's yeast extraction of S.C. at the different doses of 10, 20 and 50 mg/mL improved sperm viability and motility.

For female ruminants, it was proved that dietary supplementation with fermented yeast culture induced an early onset of the first post-partum heat, improved the conception rate and reduced the number of inseminations per conception [38].

In the same way, our results indicate that rams fed with S.C. had higher sperm motility and viability and a lower number of abnormal sperm. Based on these findings and those proved by [39,40], we could hypothesize that the antioxidant properties of S.C. could stimulate the improvement of semen characteristics. Dietary antioxidants are crucial to control and counteract the harmful effect of oxidative stress. Therefore, a sufficient antioxidant intake contributes to a lower risk of oxidative stress-mediated diseases and infertility. For that reason, yeast has been shown to contain significant amounts of antioxidants [41]. In addition, S.C. supplementation in dairy cows' diet increased total antioxidant capacity, glutathione peroxidase and superoxide dismutase in serum while decreasing malondialdehyde [42].

Furthermore, this improvement in sperm quality could be explained, in part, by the positive effect of S.C. on diet digestibility. One of the limitations that influences the use of cereal crop residues in ruminant diets, such as wheat straw as used in this study, is its low digestibility and protein deficiency [43]. Our results showed that for nutrient digestibility, the S.C. yeast supplementation tended to increase the crude protein digestibility and significantly improved the digestibility of dry matter, organic matter and crude fiber. Similar results were reported earlier for lambs [44], dairy heifers [45] and buffalos [46]. This improvement in nutrient digestibility could be due to different possible yeast action modes in a ruminal environment. It is admitted that the presence of a yeast source in the rumen improves the release of various proteolytic, glycolytic or lipolytic enzymes, which leads to better digestion of the organic matter [47] and promotes the ruminal ecosystem through the stimulation of the growth and the function of ruminal microbiota [48]. This stimulation is ensured by releasing amino acids and monosaccharides through the cell wall [49].

Our results showed a significant increase in fiber digestibility with the addition of yeast to a wheat straw-based diet for rams, which is in agreement with the findings of other authors such as Ghazanfar et al. [45] and Mallekahi et al. [44] who reported a significant positive effect of yeast on neutral detergent fiber, acid detergent fiber and cellulose digestibility. It was also shown by Ding et al. [50] that S.C. increased the number of the total ruminal bacteria, fungi and protozoa, especially the Ruminococci, by two- to four-fold [12] in fibrolytic protozoa and *Fibrobacter succinogenes* feed colonization [51]. Furthermore, in one of the few papers that studied the efficiency of yeast addition on gene expression, Durand et al. [51] reported an increase in the expression of carbohydrate-active enzymes GH5 and GH43 in the rumen of lambs receiving yeast supplementation. This suggests the ability of this feed additive to improve the fibrolytic potential of the rumen microbiota.

However, findings regarding crude protein digestibility are inconsistent. Similar to our findings, Osita et al. [52] showed that crude protein digestibility improved by 24.5% by adding 1.5 g yeast/kg to the sheep's diet. Likewise, Malekkhahi et al. [44] found that crude protein digestibility improved by 7.8% by supplementing the wheat straw-based lambs' diet with 4 g yeast/animal/day. In another study on dairy heifers, Ghazanfar et al. [45] recorded

an improvement of 6% in crude protein digestibility by adding 5 g of yeast/animal/day on a wheat straw-based diet. However, some other studies found that yeast supplementation does not significantly affect crude protein digestibility [27,46]. Therefore, as proved by the meta-analysis study of Desnoyers et al. [28], all ruminal fermentation characteristics influenced by yeast addition are significantly dependent on at least one of the dietary characteristics, such as the concentrate proportion.

## 5. Conclusions

The present study has shown that the supplementation of S.C. into the diet of rams improves the semen quality and the diet digestibility of rams when fed low-quality straw and used in mating season. These findings suggest the importance of using prebiotics in animal feeding to improve animal performance when raised under hard conditions and fed with low-quality forage.

**Author Contributions:** S.B.S.: conceived and designed the experiment, investigation, formal analysis, visualization, and writing—original draft; J.J. and S.A.: writing, software and formal analysis; A.N. and S.K.: writing—review; Z.M.: contributed new reagents (Celmanax) and methodology; M.A. and M.K.: analytical tools; M.M.: manuscript revision and funding. All authors have read and agreed to the published version of the manuscript.

**Funding:** This research was funded by the Arm & Hammer Animal and Food Production.

**Institutional Review Board Statement:** All procedures employed in this study meet ethical guidelines and adhere to Tunisian legal requirements (The Livestock Law No. 2005-95 of 18 October 2005).

**Informed Consent Statement:** Not applicable.

**Data Availability Statement:** The data presented in this study are available on request from the corresponding author.

**Acknowledgments:** We are deeply grateful to the staff of the experimental farm of ESAKef for the feeding experiment servicing and Arm & Hammer Animal and Food Production for the financial support.

**Conflicts of Interest:** The authors declare no conflict of interest.

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
