# Peer review of "Effect of Saccharomyces cerevisiae Supplementation on Reproductive Performance and Ruminal Digestibility of Queue Fine de l’Ouest Adult Rams Fed a Wheat Straw-Based Diet"

_agriculture, doi:10.3390/agriculture12081268_

Round 1
Reviewer 1 Report
Author comments
Dear authors. Thanks for consider this journal and its special number to publish your projects’ outcomes. I hope you will find constructive the following comments. There are issues needing clarification specially in the determination of feedstuff nutritional quality for ruminant nutrition.
General comments
It is important to question authors for their decision on the reported values of fiber. The CRUDE FIBER is not a value related with nutritional quality for ruminant nutrition. Instead, Neutral Detergent Fiber and Acid Detergent Fiber (NDF and ADF, respectively) are the recommended in ruminant livestock nutrition. Please clarify this decision and if necessary, modify according it.
In my opinion, the title is not properly used. Reproductive capacity is a multifactorial term involving many criteria within which sperm quality and production are part of.
Products, doses, and administration ways for vaccines and anthelmintics has to be included
Please be aware of italics use in scientific names and general acronym use. For example, the yeast’ name appears as S.C ad others SC
The introduction highlights the lacking of works related with yeast supplementation over fertility. However, the nutritional issues are not properly mentioned during this section. Thus, it is somewhat confusing.
The study duration is not properly mentioned in the material and methods section. It appears in the abstract but is not clear onwards
Authors obtained data from 6/7 different points. However, statistical analyses and tables show a comparison between the beginning and the end of three experiments. Authors are invited to clarify this methodological and statistical decision.
In the line 109 authors argued that “genera appearance of semen was visually assessed” but ther is not further information on this subject.
L290: Authors argue “Thus, these factors confirm the results of the present study”. But this should be not declared since those previous studies were done with different species and in other experimental conditions.
L76-77: The process to evaluate the healthiness of rams should be improved
L84: Please add measurements of trial pens.
L99: The process of corporal condition assessment should be referenced
L107: Please establish the process to hormonally induce ewes
L114-115: Please use a reference on the eosin/nigrosin technique
Specific comments
Title: “Digestive aspects” is difficult to understand within a title. Maybe digestibility?
Title: Please consider removing “the”
L19: nutrient digestibility
L36: Please consider omit the word “the” before “livestock”
L38: Please consider remove “proven to be”
L38: It is recommended to avoid words like “brutal”
L36 and 38: Please consider omit either “small ruminants or “ewes”
L40: “However” is not the right connector
L41: Please consider that goat is another species
L47: Has become
L54: “DM” was not defined yet
L54-55: Please define whether use “in ruminants” or “in cows”
L56: The referenced study deals only with camels
L56: Elghandour et al
L57: Reduced
L59: Few studies have focused
L62: Consider changing “autochton” for indigenous or native
L68: Please consider omit the word “and”
L74: Animals aged…
L76: Acronyms are recommended when the word is repeated some or many times amongst paragraphs.
L81: Be careful with the use of parentheses
L83: is detailed
L113: Please consider changing “determinated” for “”assessed”
L121: please put in vivo in italics
L128: ml
L126: was
L127: Please change “their” for “posterior” or “further”
L127: chemical analyses is incomplete because those posterior procedures were more than chemical
L131: Analytical analysis sounds jargon. Please consider change it
L132: “Distributed” is somewhat confusing
L135: What is AOAC
L137: Please consider eliminating the (n = 3)
Table 2: Acronyms are misspelled.
Table 2: “TD” is not defined
L160: “Semen” is in italics
L162: “is presented” …….it has to be in present
L169: what is “rose”
L171: > 0.05
L178: If P was equal to 0.05 thus it has to be considered as significant
L249_250: All the acronyms might be omitted since they only appear once or twice within the document
L249: Nutrients
L244: Authors argue that supplementation was did in milligrams but this does not appear before
L246-247: Explanation is somewhat ambiguous
L252: 0, 3, 6, and 12.6 g/day
L253: nutrient
L259: In another trial…
L267: What is MEM?
L277: were
L297: What is MDA? Acronym is not needed
L297-298: The use of such reference is somewhat wear
L302: Please consider to eliminate “digestive”
L315: Incorrect use of citation style
L319: F. succinogenes……… has to be in complete version
L321: Incorrect use of citation style
Author Response
On behalf of the co-authors, I would like to thank you for revising our manuscript.
Please see the attachement, where we put our point-by-point responses to all the comments and suggestions indicating where and how modifications to the text were implemented
Regards

Reviewer 2 Report
Dear authors, it seems to me that this manuscript has great relevance in the scientific world. However, important points affect the quality of the manuscript.
The abstract is not complete. Add the average age and experimental design. Remember, the abstract is the document that first represents your manuscript. Improve the conclusion of the abstract.
The introduction is very generic and when you read it, there is a need to reread it. This is because some points are confusing or there is a lack of connection between ideas. Also, comparing the title, abstract and introduction, I can say that the aim of the manuscript is not clear.
Methodological aspects decrease the quality of the manuscript: number of experimental units; age of the animals, complete description of the diet, description of the feeding management, bromatological composition (crude fiber? Really?), etc.
Author Response

(The authors gave the same response as above.)

Reviewer 3 Report
This paper investigates the effect of dietary Saccharomyces cerevisiae (S.C) supplementation on sperm quality, nutrients digestibility and nitrogen balance, providing rational for the potential application of S.C in rams feed. This is an interesting and useful work, but some aspects remain too preliminar and descriptive. The authors speculate that the reason for the improvement of semen characteristics could be attributed to the antioxidant property of S.C, but it is a pity that no data about antioxidant capacity is provided to support this claim. Remaining experiments performed in this manuscript to my opinion remain descriptive, and do not provide the proposed mechanistic understanding. In addition, there are some textual errors, and some references are missing and should be added. Detailed comments are as follows:
Sometimes genus/species is italicized when they should all be.
Please italicize P (statistics) throughout.
Line 21. kg/d
Line 25. Please review the concepts of digestibility in this paper. May be you measured apparent digestibility, and not necessarily digestibility.
Line 79. Please provide the feeding level of concentrate.
Line 99. Describe body score condition measurement or provide reference.
Line 124. Were urine and feces samples pooled within this 7-d periods? Clarify.
LIne 217. Please find a uniform font type or size for this figure..
Line 257-259. This is another study. Please revise this sentence to avoid any ambiguity.
Line 266-268 Please provide references.
Line 322 What is CAZymes CH43?
Author Response

(The authors gave the same response as above.)

Reviewer 4 Report
The manuscript evaluates the effects of dietary Saccharomyces cerevisiae on feed intake, nutrients digestibility, nitrogen balance, body weight and sperm quality of mature Queue Fine de l’Ouest rams. The material and methods are sufficiently detailed, the results are presented in tables and figures and provide all essential information that was well interpreted and discussed. Except for some typos the text is well written, it is an interesting manuscript and gives relevant information to researchers and farmers working with this breed, however, this could restrict the number of authors quoting this paper. In general, the manuscript has scientific merit, ethical acceptability and meets the journal's standard, so I believe it is suitable for publication after some revisions (please see below some specific comments and suggestions).
Specific comments
Line 2 – Saccharomyces cerevisiae in italics.
Line 3 – Remove “the” before adult
Line 27 – Remove “one”
Line 29 – Remove “one”
Line 30 – “recommend” is such a strong word. Maybe you can use something like “indicate the potential of SC in improving semen quality of mature rams fed with low quality forage”
Line 101 – Remove “)” after diameter
Line 171 – Correct the p-value “P>0.05”
Line 184 – Rearrange the sentence or the values inside the parentheses “26.8 ± 3.85 and 9.28 ± 0.95%” (the way it is the values refers to control and SC respectively).
Line 226 – replace “NU” with “UR”
Line 227 – replace “NF” with “FN”
Line 244 – S. Cerevisiae in italics
Line 247 – This may be explained in part by the fact that there was not any measured difference in the feed intake between both dietary groups. I think another important consideration to be made is the age of the animals (already mature)
Line 252 – … at increasing addition rates “of 0, 3, 6 and 12.6 g/day”
Line 253 – Remove “study”
Line 254 – “proved” is such a strong word
Line 257 – Remove “proportion in the diet”
Line 257-260 – Rewrite that sentence… which study?
Line 280 – I think you can stick to references with ruminants. Consider removing this sentence
Lines 285-286 – I think you can stick to references with ruminants. Consider removing this sentence
Lines 297-298 – I think you can stick to references with ruminants. Consider removing this sentence
Line 310 – Replace “organic matter” with “OM”
Line 317 – S. cerevisiae in italics
Line 345 – Consider using other words than “bad conditions” and remove “of local”
Line 349 – Remove “and.”
Tables – Try to standardize the information like whether or not to present the p value in all tables and describe what "beginning” and "end" refers to
Figure 1 – Again, try to standardize the information: use the same text font and legend to indicate the treatments (see that Figure 1a uses a different legend than Figure 1d), sometimes you use “collection rank” and then “collect rank”
Author Response

(The authors gave the same response as above.)

Reviewer 5 Report
I have only two comments
The article needs intensive language editing
Secondly, all statistical analysis of the manuscript wrong, how such design that contains control and treated group (two independent groups) and must be analysed by independent T test analysis here you analysed the whole experiment but ANOVA which is available only with more than two groups
Author Response

(The authors gave the same response as above.)

Round 2
Reviewer 1 Report
L326: acronyms for DMI were not previously defined
L335: The phrase “However, no differences were observed in sperm viability and scrotal circumference” has no sense as located within the discussion section.
There are typos in L339 and L371
References in L363, 398 and 404 are incorrect.
Emmanuel (L352) is referenced as a single author which is incorrect
In line 366 authors associated outcomes to a higher feed intake, however this study demonstrated no changes in intake between experimental groups.
Thanks for allowing me to review this manuscript and I hope both authors and editor can find the most appropriated resolution
Author Response
Please find below author response to your comment.
Point 1: L326: acronyms for DMI were not previously defined: Revised accordingly
Point 2: L335: The phrase “However, no differences were observed in sperm viability and scrotal circumference” has no sense as located within the discussion section. The sentence has been removed
Point 3: There are typos in L339 and L371: Revised accordingly
Point 4: References in L363, 398 and 404 are incorrect. Revised accordingly
Point 5: Emmanuel (L352) is referenced as a single author which is incorrect: Revised accordingly
Point 6: In line 366 authors associated outcomes to a higher feed intake, however this study demonstrated no changes in intake between experimental groups. The idea has been revised and the sentence has been rewritten
The authors would like to thank the reviewer for revising the manuscript!

Reviewer 2 Report
Dear authors, I congratulate you on the work you have done. You made corrections to the manuscript according to my suggestions. I agree with the changes and believe the manuscript can be published as is.
Author Response
Dear authors, I congratulate you on the work you have done. You made corrections to the manuscript according to my suggestions. I agree with the changes and believe the manuscript can be published as is.: The authors would like to thank the reviewer for manuscript revision
Reviewer 3 Report
There are a number of minor textual errors, e.g.
Line 45. Kg/d.
Table 4. 60.3..
Line 398. Wrong cite format.
Line 404. Wrong cite format.
Line 406. It should be GH43 and not CH43.
Line 423. Wrong sentence.
The authors should give the writing a close look to expunge similar errors that I'm sure I've missed.
Author Response
Point 1: Line 45. Kg/d. Revised accordingly
Table 4. 60.3.: Revised accordingly
Line 398. Wrong cite format. Revised accordingly
Line 404. Wrong cite format. Revised accordingly
Line 406. It should be GH43 and not CH43. Revised accordingly
Line 423. Wrong sentence. Revised accordingly
The authors should give the writing a close look to expunge similar errors that I'm sure I've missed. : the manuscript has been verified
The authors would like to thank the reviewer for manuscript revising !

Reviewer 5 Report
Authors respond well to the comments but the article still needs some language editing with professional organization.
Author Response
Point 1: Authors respond well to the comments but the article still needs some language editing with professional organization: The authors would like to thank the reviewer. The manuscript has been revised by a professional organization as requested